# A Comparison of the Genotoxic Effects of Gold Nanoparticles Functionalized with Seven Different Ligands in Cultured Human Hepatocellular Carcinoma Cells

**DOI:** 10.3390/nano12071126

**Published:** 2022-03-29

**Authors:** Danielle Mulder, Cornelius Johannes Francois Taute, Mari van Wyk, Pieter J. Pretorius

**Affiliations:** Human Metabolomics, Potchefstroom Campus, North-West University, Potchefstroom 2351, South Africa; ftaute2015@gmail.com (C.J.F.T.); 12791733@nwu.ac.za (M.v.W.); pietp484@gmail.com (P.J.P.)

**Keywords:** biofunctionalization, comet assay, cytotoxicity, genotoxicity, gold nanoparticles, HepG2

## Abstract

Gold nanoparticles (GNPs) have shown great potential in diagnostic and therapeutic applications in diseases, such as cancer. Despite GNP versatility, there is conflicting data regarding the toxicity of their overall functionalization chemistry for improved biocompatibility. This study aimed to determine the possible genotoxic effects of functionalized GNPs in Human hepatocellular carcinoma (HepG2) cells. GNPs were synthesized and biofunctionalized with seven common molecules used for biomedical applications. These ligands were bovine serum albumin (BSA), poly(sodium 4-styrene sulfonate) (PSSNA), trisodium citrate (citrate), mercaptoundecanoic acid (MUA), glutathione (GSH), polyvinylpyrrolidone (PVP), and polyethylene glycol (PEG). Before in vitro genotoxicity assessment, inductively coupled plasma mass spectrometry was used to determine GNP cellular internalization quantitatively, followed by cell-based assays; WST-1 to find IC 30 and ApoPercentage for apoptotic induction time-points. The effect of the GNPs on cell growth in real-time was determined by using xCELLigence, followed by a comet assay for genotoxicity determination. The HepG2 cells experienced genotoxicity for all GNP ligands; however, they were able to initiate repair mechanisms and recover DNA damage, except for two functionalization chemistries.

## 1. Introduction

Gold nanoparticles (GNPs) are revolutionizing the discipline of nanomedicine by introducing new ways in which diagnosis, treatment, and preventative measures are carried out [1,2]. As a result of their unique chemical and physicochemical properties, such as surface plasmon resonance in the visible range, colorimetric, high electric conductivity, non-linear optical properties, high surface area, biocompatibility, and their ability to bind to a variety of ligands (coating agents), GNPs are being utilized in diverse applications [3,4,5,6]. These applications include cancer treatment, radiotherapy, drug delivery, biological imaging, diagnostic assays, chemical sensing, thermal ablation, and biosensors [7]. Ways in which GNPS has overcome drug delivery challenges include improving drug bioavailability and aqueous solubility. They have reduced toxic drug side effects by preventing drug degradation and prolonging drug release. They have also provided a platform for targeted delivery, allowing easy administration [8]. Current cases with the Food and Drug Administration (FDA), which are either undergoing investigating or approved cancer therapeutics which are GNP based, are tabulated in [9].

Despite GNP benefits, toxicity studies of how these particles may impact human health are largely limited [10,11,12]. GNPs are considered relatively biologically inert. Depending on their size, they are excreted via the urine or accumulate in the liver. There are many in vivo studies that have indicated how nanomaterials induce toxicity. However, these studies are limited by short time intervals and fail to consider the regeneration abilities of the liver. An example of GNPs causing toxicity in hepatocytes includes excessive production of free radicals and reactive oxygen species (ROS), which results in carbonylation of cellular proteins, DNA damage, and lipid peroxidation, with more devastation linked to the smaller sizes. [13,14,15,16,17,18]. One study showed that GNPs induce toxicity by binding to the DNA grooves [19]. Some of the published genotoxicity and cytotoxicity results are conflicting. Areas of discrepancy include oxidative stress and cytotoxicity [20,21]. Factors contributing to these conflicting conclusions could be GNP coating, size, physicochemical characteristics, and cell lines [22]. An example of this would be 13 nm citrate capped GNPs were found toxic in a human carcinoma lung cell line but not in a human carcinoma liver cell line at the same concentration [23]. These conflicting conclusions have led to the global drive in developing standardized testing procedures for risk assessment, led by the Organization for Economic Cooperation and Development (OECD) [10].

This study aimed to determine the possible genotoxic effects of functionalized GNPs in Human hepatocellular carcinoma (HepG2) cells. Small molecules and (bio)polymers commonly used for surface functionalization, found in the literature, were explored. The functionalized GNPs were evaluated for chemical stability using various buffers to mimic physiological and extreme conditions. The IC 30 of the GNPs was determined by using a WST-1 cell proliferation assay, followed by a cell-based assay normalized towards quantitative percentage internalization. xCELLigence was used to assess the effect of the IC 30 on cell growth in real-time. ApoPercentage experiments were used to find the possible highest toxicity induction time upon chronic exposure to the cells, which was also used to determine the genotoxic effect of the GNPs to the cells [22].

## 2. Materials and Methods

Hydrochloroauric acid (HAuCl_4_), trisodium citrate, sodium chloride (NaCl), Bovine serum albumin (BSA), poly(sodium 4-styrene sulfonate) (PSSNA), trisodium citrate (citrate), mercaptoundecanoic acid (MUA), glutathione (GSH), polyvinylpyrrolidone (PVP molecular weight 10,000), polyethyleneglycol (PEG), hydrochloric acid (HCl), glucose, 1 × phosphate-buffered saline (PBS) at pH 7.4, sodium bicarbonate (NaCO_3_), ethylenediaminetetraacetic acid (EDTA), 10 mM 3-morpholinopropane-1-sulfonic acid (MOPS pH 9 and pH 10), 100 mM 2-[4-(2-hydroxyethyl)piperazin-1-yl]ethanesulfonic acid (HEPES pH 7 and pH 8), 1 × phosphate buffer saline (PBS) buffer, 5 mM citrate (pH 4 and pH 5), 20 mM glycine (pH 2, pH 3, pH 9 and pH 10), β-mercaptoethanol, WST-1 cell viability kit, were all purchased from Sigma-Aldrich, Johannesburg, South Africa. Complete cell culture medium (Gibco Roswell Park Memorial Institute (RPMI) with fetal bovine serum), Value fetal bovine serum (FBS), and Penicillin-streptomycin (10,000 U/mL) were all purchased from ThermoFisher Scientific, Johannesburg, South Africa. The ApoPercentage assay kit was purchased from BioColor, Carrickfergus, UK.

Figure 1 illustrates the work flow of the various methods used in the study for simplicity purposes. 

### 2.1. Gold Nanoparticle Work

#### 2.1.1. Gold Nanoparticle Synthesis Using the Turkevich Method 

All the glassware was immersed in aqua regia, followed by ultrapure Milli-Q water (18.2 MΩ cm^−1^) rinsing. A rapidly stirring 100 mL 0.25 mM HAuCl_4(aq)_ was heated to boiling in a 500 mL Erlenmeyer flask on a Stuart stirrer-hotplate. Upon reaching boiling point, 2300 µL 1% trisodium citrate (aq) was added all at once. The reaction was left to proceed for approximately 15 min until the solution turned a rose-red color. The suspension was then placed in an ice bath to cool down. The GNP suspension was then filtered using a 0.8/0.2 µm Super membrane sterile filter (Pall Life Sciences, Port Washington, NY, USA) and kept at 4 °C until needed [24].

#### 2.1.2. Functionalization by Ligand Exchange

The seven ligands (coating agents) for functionalization were chosen based on their structural differences and biological applications. These ligands were bovine serum albumin (BSA), poly(sodium 4-styrene sulfonate) (PSSNA), trisodium citrate (citrate), mercaptoundecanoic acid (MUA), glutathione (GSH), polyvinylpyrrolidone (PVP), and polyethylene glycol (PEG). The ligands represented altered surface charges, molecular weights, and biocompatibility (Appendix A).

The GNP suspension was divided equally for ligand exchange in 15 mL Falcon tubes. Each GNP aliquot was pH adjusted (NaOH, NaCO_3,_ or HCl) to the corresponding ligands isoelectric point (Appendix A). The ligands were added to the GNP suspension and vortexed for 30 s. The experiment control was Milli-Q water (18.2 MΩ cm^−1^) added to an aliquot of GNPs and treated exactly like the other samples. The GNP-ligand mixtures were left at room temperature overnight. The excess ligand was removed by centrifugation at 1000× *g* (Hermle Z 206 A, fixed angle rotor, Lasec, Cape Town, South Africa) for 90 min. The supernatant was removed, and the pellet was re-suspended in half the original sample volume to concentrate the samples.

#### 2.1.3. GNP Characterization and Stability

Ultraviolet-visible spectroscopy (UV-Vis spectrum) was used to determine GNP-citrate size, concentration, and morphology (control particles), ligand functionalization success, and assess GNP-ligand stability in similar conditions to those found in cell culture. High-resolution transmission electron microscopy (HR-TEM) was done on the GNP-citrate as the “control” particle to ensure that spherical nanoparticles were successfully synthesized. The particle distribution was also determined with the GNP-citrate particles. Dynamic light scattering (DLS) was used to determine the hydrodynamic diameter of the GNPs once they were functionalized. Relative functionalized GNP-ligand charge was determined using agarose gel electrophoresis.

#### 2.1.4. GNP Morphology and Size Distribution

##### UV-Vis Spectrometry

UV-Vis spectra for all the GNP-ligands were obtained using an HT Synergy microplate reader (BioTEK, Winooski, VT, USA) UV-Vis spectrometer (200–1000 nm) with Gen 5.1 as the corresponding software. A 50 µL aliquot of the GNP-ligands was transferred to a 96-well micro-well plate. The spectral range chosen was 350–800 nm, following the literature [25], where a blank reading was obtained using ddH_2_ O (18.2 MΩ cm^−1^). The maximum absorbance peak, also known as the surface plasmon resonance peak (λ_SPR_) and the absorbance at 450 nm, was used to estimate the size and concentration of the citrate-capped GNPs as the control particles. The values were calculated using the method described by Haiss et al. [25]. UV-vis was also used to determine functionalization success by noting a change in the UV-Vis spectra compared to the GNP-citrate spectrum.

##### High-Resolution Transmission Electron Microscopy (HR-TEM)

HR-TEM was used to look at the morphology of the GNP-citrate to ensure that spherical GNPs were synthesized correctly. Micrographs were obtained using a Tecnai F20 high-resolution field emission transmission electron microscope (ThermoFisher, Gauteng, South Africa). Morphology and size dispersity was determined using Image J V1.46 r. This method only considers the core of the particle.

##### Dynamic Light Scattering (DLS)

DLS determined the hydrodynamic diameter of all the prepared GNP-ligands. This diameter includes the ligand where HR-TEM only examines the particle core. DLS was done using the Zetasizer Nano (Malvern, UK) module using (Zetasizer version 6.20 software) and polystyrene cuvettes. The polystyrene cuvettes were prepared by being rinsed three times with Milli-Q water and inverted on a paper towel to dry overnight. Once the cuvettes were dry, they were stoppered until needed, preventing dust interference. A dilution series of the GNP-ligands was prepared to optimize the assay. It was found that for each GNP-ligand, a volume of 63 µL GNP-ligand added to 937 µL ddH_2_O was optimal for the suspensions.

##### Biofunctionalized GNP Net Charge

The charge estimation of the functional group was determined by performing an agarose gel electrophoresis assay. GNP samples were electrophoresed (50 volts, 033 mA, Baygene, BG-power 300, (Vacutec, Roodepoort, South Africa)) for 30 min using 1 × TBE (Tris-Borate-EDTA) or 1 × TAE (Tris-Acetate-EDTA) buffer in a 0.25% agarose gel. The loading buffer was 20% *v/v* glycerol_(aq)_ (80% *v/v*) and 80% *v/v* GNP-ligand.

#### 2.1.5. GNP-Ligand Stability Test

The various parameters investigated were typically used biological buffers of different ionic strengths, pH ranges, and a mammalian tissue culture medium (with fetal bovine serum). These parameters included 1 mM salt (NaCl), 0.01 × EDTA, 10 mM MOPS (pH 9 and 10), 100 mM HEPES (pH 7 and 8), 1 × PBS buffer and complete (RPMI with fetal bovine serum) cell culture medium. The organic molecules investigated included 5 mM citrate (pH 4 and 5) and 20 mM glycine (pH 2, 3, 9, and 10); 20 µL β-Mercaptoethanol was used to represent thiol-containing compounds; 30 µL particles were added to a 96-well plate for UV-Vis spectrometry for each GNP-ligand. Readings were obtained immediately upon adding the solution to the GNP-ligand, with subsequent readings at 1 h, 2 h, 6 h, 12 h, and 24 h. The particles were considered stable if the maximum optical density of the UV-Vis decreased by less than 70% at that specific time-point.

### 2.2. Cell Culture Work

#### 2.2.1. Human Hepatocellular Carcinoma HEPG_2_ ([HepG2] ATCC HB−8065) Cell Culturing

HepG2 cells were chosen as they are a commonly used cell line in literature for nanotoxicity studies.

##### General Cell Culturing of the Cell Line

Sterile techniques were used at all times when the cells were handled. Cells were retrieved from −80 °C cryostorage freezer and thawed according to ATCC recommendations (American Type Culture Collection, non-profit, global biological resource center). They were seeded in a T25 (25 cm^2^) (Corning, Axygen, Sigma-Aldrich, Johannesburg, South Africa) vented cap flask containing 10 mL RMPI 1640 medium supplemented with a final concentration of 10% FBS and 1% Penicillin-streptomycin, cultured using a 37 °C 5% CO_2_ incubator (HeraCELL, Heraeus, ThermoFisher, Johannesburg, South Africa). After 16 h, the cells were rinsed with 1 × PBS to remove the DMSO, and 10 mL fresh medium was added to the cells.

##### Trypsinization and Cell Counting

The cells were trypsinized upon reaching 70–80% confluency. The cells were rinsed twice with 1 × PBS; 1 mL 1 × trypsin was added, followed by incubation at 37 °C for 5 min. The cells were dislodged from the T25 flask by gentle hand palm tapping, after which 2 mL medium was added. The contents of the T25 flask were transferred to a 15 mL Falcon centrifuge tube to sediment the cells and remove the trypsin-medium at 1000× *g* for 5 min. The cell pellet was suspended in supplemented medium to a final volume of 10 mL. 

Cell counting was done using a handheld cell counter (Merck, South Africa). In a 1.5 mL microtube, 900 μL 1 × PBS and 100 μL cell suspension were mixed. The cells were then counted. All cell experiments were performed at passage 28. Regular analysis for the presence of mycoplasma was also conducted.

#### 2.2.2. Cell-Based Assays to Determine Cytotoxicity

##### WST-1 Cell Viability Kit

The WST-1 cell viability kit (Sigma-Aldrich, Johannesburg, South Africa) was used to determine the cytotoxicity of the GNP-ligands to HepG2 cells over a 24 h period, including establishing the GNP-ligand inhibition concentration of 30% (IC 30). The IC 30 was chosen to allow 70% of the cells to remain viable after chronic exposure without causing excessive cellular stress responses. The IC 30 would also allow for ascertaining GNP-ligand genotoxic effects [26]. The wells of a 96-well plate were seeded with 7500 cells/0.32 cm^2^ in 100 µL medium and incubated overnight in culturing conditions to allow for cellular attachment (this cell density was used for all the following methods). The next day, the spent medium was removed, cells were rinsed twice with sterile 1 × PBS and supplemented with fresh 100 µL medium. The GNP-ligands, controls, and blanks were added to the cells to a final volume of 200 µL/well followed by 24 h incubation. The five concentrations had final ligand concentrations of 1710 pM, 860 pM, 430 pM, 220 pM, and 100 pM, respectively. These calculations assume that all the ligands were functionalized onto the GNP surface. This method had three controls. The positive cell control contained cells with medium and water only. The negative cell control contained cells exposed to 5 mM hydrogen peroxide and a ligand control containing cells exposed to a 2200 pM concentration of ligand only for each ligand (no GNP present).

After 24 h treatment, the media was removed. Cells were washed twice with PBS, and 100 µL fresh medium was added, with the addition of 10 µL WST-1 reagent. The 96-well plate was then placed in the spectrophotometer, shaken for 10 s, and incubated at 37 °C for 30 min. Optical density readings were obtained at 460 nm (reference wavelength was 660 nm). The Gen 5.1 software allowed for an automated blank subtraction. This test was done in triplicates. The % cytotoxicity was calculated as follows [27]:(1)$$\%\;\mathrm{cytotoxicity}:\frac{\left(\mathrm{Cell}\;\mathrm{Control}\;–\mathrm{experimental}\right)}{\mathrm{Cell}\;\mathrm{Control}}\times \;100$$

Cell control = untreated cells in medium and water.

The cells, PBS washes, and culture media used in the WST-1 assay were then taken for ICP-MS analysis.

##### Determination of GNP-Ligand Cellular Internalization Using Inductively Coupled Plasma Mass Spectrometry (ICP-MS)

ICP-MS is a technique that can measure elements at trace levels in biological fluids. In this case, it was used to determine whether the GNP-ligands entered the cells, attached to the surface (sediment), or had no cellular interaction. The HepG2 cells used in the WST-1 assay were digested with 70% nitric acid overnight, diluted to 1% nitric acid concentration, and then analyzed. The apparatus used was the Agilent 7500 CE series ICP-MS, radio frequency (RF) power of 1550 W, sample depth of 8 mm. The carrier gas was 0.92 L/min, makeup gas was 0.25 L/min, nebulizer contained MicroMist, the spray chamber temp was 2 °C, with an integration time of 100 ms.

##### Apoptosis Induction Time-Point

The ApoPercentage assay kit (BioColor, Carrickfergus, UK) was used to determine whether the GNP-ligand was causing necrosis or apoptosis and to calculate the induction time-point of apoptosis, using the IC 30 as determined by the WST-1 assay (Appendix A). If the cells undergo necrosis, then the GNP-ligands are causing physical cell damage, causing mechanical rupture, and further analysis of the GNP effect cannot be determined. The apoptotic induction time-point is where apoptosis is at its highest and where the biochemical process in the cell changes in response to the GNP-ligand concentration. This is the time-point that would be used to determine the DNA damage [28].

Cells were seeded into a 96-well plate and allowed to attach. A time-dose experiment was done for the 0 h, 3 h, 6 h, 12 h, and 24 h time points. The assay kit manual was followed but adapted to a 96-well microplate format. At the zero-hour time-point, the GNP-ligands were added to the medium and removed immediately from the cells; 95 µL medium mixed with 5 µL dye was added to the cells and incubated at 37 °C for 30 min, after which the cells were rinsed with 200 µL 1 × PBS. For microscope visualizing, 50 µL 1 × PBS was added to each well, and micrographs were taken of the cells using a standard camera (Leica DMIL 40× magnification). Fifty microliters of dye release solution (ApoPercentage kit, Biocolor life science assays, Carrickfergus, UK) were then added to the wells. The plate was shaken for 10 min in the spectrophotometer and then read at 550 nm. The same procedure was followed 3 h, 6 h, 12 h, and 24 h later.

##### Real-Time Cellular Growth and Cytotoxicity Determination

xCELLigence technology was used to determine cell growth and cytotoxicity of the GNP-ligand to the cells in real-time. The resistor plate was set up in the RTCA SP machine (Roche, South Africa) as per the manufacturer’s (RTCA software manual version 1.2) instructions. Step 1 was the 100 µL medium background check. Step 2 was cellular addition (WST-1 seeding method with 16 h cell attachment), with Step 3 being the addition of the IC 30 GNP-ligand concentration. The machine was left to run for 24 h, and then the program was terminated.

The software was set up according to the following parameters: 

Resistor plate read: The resistor plate was set up in the RTCA SP machine; (Roche) as per manufacturer’s (RTCA software manual version 1.2) instructions.

Program set-up: Under the layout tab—All the wells utilized were highlighted, and “Apply” was selected. The sample identification was entered under “compound name.” Under the schedule tab—Step_1 was the medium background check. The set-up for this was 5 sweeps and 1-min intervals. Step_2 was the addition of the cells. The set-up for this was 20 sweeps and 3 min intervals. The sub-step was 999 sweeps and 10 min intervals. Step_3 was the addition of the gold nanoparticles. The same set-up was used as in Step_2. The machine was left to run for 24 h, and then the program was terminated.

Plate set-up: To the wells being used, 100 µL medium was added, and step_1 of the program was executed. The 96-well xCELLigence plate was seeded the same way as for the WST-1 assay, after which Step_2 was performed. The cells were given approximately 16 h to attach before they were treated with the different GNPs, at IC 30 concentration (determined via the WST-1 assay), after which step_3 was executed.

Cytotoxicity was calculated using the formula previously mentioned after subtracting the GNP reference. The negative control = cells treated with medium and water only, and the positive control = Hydrogen peroxide + medium

### 2.3. Genotoxicity Determination

The comet assay is a classical method used to determine DNA damage in single cells by gel electrophoresis [29]. The head of the comet is tightly packed DNA, whereas the tail of the comet consists of relaxed and fragmented DNA. The head versus tail fluorescence ratio determines the degree of DNA damage. The damaged DNA includes relaxed DNA double-stranded and single-stranded nicks [30]. This assay was modified and implemented [29,31].

#### 2.3.1. Comet Assay

This assay was modified from published literature [29,31].

Sample preparation: Cells were grown and sub-cultured into a 24-well plate then treated with the IC 30 concentrations of the respective GNP-ligands. An hour recovery time was implemented before the slide preparation step. This recovery time reduces the possible adverse effects on the cell handling procedure, which may skew the results. A pitfall of this recovery time is that if the GNP-ligand did cause immediate DNA damage (at 0 h), the cells could repair it during that incubation time; however, the focus was on the longer-term damage that survived the incubation time. The comet assay was conducted at 0 h, 3 h, and 24 h time-points. The 0 h was to see the DNA damage after the recovery step. The 3 h was the average apoptosis induction time-point, and the 24 h time-point was to determine the DNA damage after 24 h GNP-ligand exposure. The negative control contained medium only; 60 µM H_2_O_2_ was added 30 min before trypsinization for the positive control. The treatment followed the WST-1 procedure. After trypsinization, the samples were neutralized with 450 µL supplemented medium, which was transferred into autoclaved 1.5 mL microtubes, and placed back in the humidified incubator for the hour recovery time. This process was repeated in triplicates with an overall number of 50 comets per time point for each GNP-ligand.

Slide preparation: The slides were coated with 300 µL 1% high melting point agarose and then left to set. Once the slides had set, they were then inserted into the comet assay manifolds, ready for the addition of the cells. Approximately 20 cells per well were added into the wells of the manifolds. To 100 µL 0.5% low melting point agarose, 50 µL cell mixture was added, the gel and cells were mixed gently together, and then of the cell/gel mixture, 20 µL was added to a manifold well. The station was placed on ice allowing the gel time to set. Once the gel had set, the slides were carefully removed from the manifolds ensuring that no gels were detached from the slide.

Comet assay procedure: The slides were then placed into lysis buffer for 16–20 h at 4 °C, after which they were then transferred into the electrophoresis buffer and incubated for 30 min at 4 °C. The electrophoresis power pack (Bio-Rad, South Africa) was switched on and left to run at 40 volts for 30 min at 4 °C. Once this was done, the slides were then transferred into Tris-HCl (pH 7.4) buffer for 15 min at 4 °C, after which they were then transferred into ethidium bromide staining solution and stained for 30 min at 4 °C. The excess stain was then rinsed off by incubating the slides in Milli-Q water for 5 min at room temperature. The comets were then viewed at 20× magnification on an Olympus lX70 light microscope fitted with a green filter. The software used to score the comets was Comet Assay IV (Perceptive Instruments, Bury Saint Edmunds, UK). The data was analyzed based on the tail intensity data generated by the software.

#### 2.3.2. Statistical Analysis for Comet Assay (DNA Damage Statistics)

The significance of the DNA damage was determined by using statistics. Welch ANOVA tests were performed for each ligand to assess the significance of damage across the three time points. An analysis of variance or ANOVA was performed instead of a *t*-test when more than two groups were compared. The traditional one-way ANOVA assumes equal group variances, however, since this assumption did not hold based on Levene’s test for equal variance [32]. Welch’s modified ANOVA [33] is robust against violations of the assumption of homoscedasticity. The Games–Howell test was used as a post hoc test to assess pairwise significance (indicated by the asterisks). It is also robust against violations of the assumption of homoscedasticity [32].

## 3. Results

### 3.1. Gold Particle Synthesis and Stability Evaluation

The GNP synthesis and ligand exchange processes were successfully carried out (Figure 2). They were characterized with HR-TEM (Figure 2a), UV-Vis (Figure 2c), and DLS (Table 1). HR-TEM visually showed that the GNPs synthesized were morphologically spherical (Figure 2a). Image J analysis of the particle distribution for 230 analyzed particles indicated that they were predominantly ±18 nm in diameter (Figure 2b). The diameter and concentration of the GNPs were calculated as ±18 nm and 1.99 nM, respectively, using the approach and equation from Haiss et al. [25]. The bell curve shape and absorbance peak at 520 nm in the UV-Vis spectra of the functionalized GNPs is a known trait for spherical nanoparticles. The shape of the curve also hints at the GNP stability after functionalization [25,34].

The results obtained for the gel electrophoresis showed that most of the functionalized particles had a negative net charge, with PEG being neutral (Appendix A). The various migration patterns of the samples were also an indication that the GNPs were successfully functionalized and had different net charges.

The GNP-ligands were similar in size, including the hydrodynamic diameter as seen with the DLS data (Summarised in Table 1). The hydrodynamic diameter considers the ligand, whereas the size estimated by the HR-TEM only considers the particle’s core.

In the stability assay, all the GNP-ligands were stable in cell culture conditions (Appendix A is a summary of the GNP-ligands that were stable in the various conditions. All the spectra can be found in the repository).

### 3.2. Cytotoxicity

#### 3.2.1. WST-1 Assay

The results obtained from the WST-1 assay are given in Figure 3. GNP-BSA, GNP-PSSNA, GNP-MUA, and GNP-PVP showed the highest concentration having a higher cytotoxicity percentage. GNP-PEG had a horizontal trend, where all the concentrations resulted in approximately the same level of cytotoxicity. GNP-GSH showed the lower concentrations having higher toxicity. Only this ligand had this trend. Attaching the ligand to GNP reduced the cytotoxic effect of the GNP-citrate, GNP-PEG, and GNP-PSSNA. A decrease in cytotoxicity percentage was seen between the ligand only control and the highest concentration percentage. Attaching PVP to the GNP increased cytotoxicity as the cytotoxicity percentage doubled from the ligand-only control compared with the highest GNP-PVP concentration. A large cytotoxic increase was observed with the GNP-BSA compared with the unbound ligand treatment.

#### 3.2.2. ApoPercentage

Fold increase intensity was used to determine the apoptotic induction time-point due to the GNP-ligands (graphs can be found in Appendix A). Table 2 contains all the apoptotic induction time points. Apoptosis was also seen in the negative control since it is a naturally occurring process. The average apoptotic induction time-point for the GNP-ligands was 3 h. The comet assay analysis evaluated the 0 h, 3 h, and 24 h exposure time-points for DNA damage.

Table 2 is a combined table of the data obtained for the three cytotoxic parameters analyzed.

The WST-1 value in Table 2 was the IC 30 concentration. The cells, PBS wash, and medium during dosing were analyzed using the ICP-MS method. Most of the GNP-ligands were present in the media, with the particles’ internalization being low except for GNP-GSH, which had a very high internalization of 92%. The normalized IC 30 was calculated based on the percentage of the GNP-ligands internalized at the specific IC 30 concentration. Fold change in cytotoxicity/relative to cytotoxicity considers the cytotoxicity of the GNP-ligand concentration based on the percentage of GNP-ligands internalized. GNP-BSA had the highest perceived toxicity of 47.8 despite the low internalization percentage. The perceived toxicity of GNP-MUA seemed to be equivalent to the amount internalized. GNP-citrate, GNP-PEG, GNP-GSH, GNP-PSSNA, and GNP-PVP all had low perceived toxicity. Once the internalization of the GNP-ligands was determined, it was imperative to know whether the occurring cell death was due to necrosis or apoptosis. The stained cells in the ApoPercentage assay indicated that the cells did undergo apoptosis and not necrosis, and therefore genotoxicity testing could proceed.

#### 3.2.3. xCELLigence

The incline in the growth curve (Figure 4a) indicated cell viability. The hydrogen peroxide control showed cell death, and none of the test samples followed this pattern. All the GNP-ligands followed the same pattern seen by the GNP-PEG and GNP-PSSNA samples. GNP-GSH was the only sample that followed the cells and water control trend. These results correlated with the WST-1 data as the chosen IC 30 concentration did not result in a drastic decrease in cell viability but rather continued to thrive. Figure 4b is the inverse of Figure 4a, and all the lines have a downward trend. There appeared to be three groups formed at this point. The first group had a gradual decline (GNP-GSH). The second group (GNP-PSSNA) had a sharper decline, with the majority of the GNP-ligands following this pattern. The third group included GNP-citrate, which had lower toxicity with a sharp decline. The downward trend of the lines (Figure 4b) was an indication that the cells were recovering from the cytotoxicity after GNP-ligand exposure. Cells exposed to GNP-GSH recovered more slowly than the other GNP-ligands. GNP-BSA was not visible on the graph as the cells recovered almost immediately (±3min). The cells exposed to the remaining GNP-ligands recovered at approximately the same rate. Normalizing the data to a single time-point enabled the determination of recovery patterns in Figure 4 [35].

### 3.3. Genotoxicity

Single-cell gel electrophoresis (Comet assay):

The average (*n* = 0) tail intensity results for 50 comets obtained for each time point and GNP-ligand from the comet assay are seen in Figure 5.

At the 3 h time-point, GNP-PVP had the greatest DNA damage, and GNP-PEG had the lowest. GNP-MUA had the greatest DNA damage at the 24 h time-point. The average tail intensity data were then subdivided into four groups of DNA damage (<6%, 6.1–17%, 17.1–35%, 35.1–60% and >60%) for each GNP-ligand and time-point. The overall DNA damage percentage (which was based on damage greater than 6%) induced by the GNP-ligands and the statistical significance are shown in Table 3.

Table 3, cells exposed to GNP-PVP resulted in 48% DNA damage at 3 h, and the cells exposed to GNP-MUA resulted in 46% damage at 24 h. This damage was statistically significant between 0 h and 24 h. The cells exposed to GNP-BSA, GNP-PSSNA, GNP-PVP, and GNP-GSH repaired the damaged DNA caused by the GNP-ligands, as the damage percentage increased from 0 h to 3 h and then decreased again at 24 h. GNP-MUA, GNP-PEG, and GNP-citrate DNA damage were not repaired at 24 h, with statistical significance seen for GNP-MUA and GNP-PEG. The negative control was cells in media only with no GNP-ligand exposure. The DNA damage seen could be attributed to cell handling or natural occurrence in the cell line [36].

Table 4 is a collective table of the information obtained for the GNP-ligands over the various parameters examined.

## 4. Discussion

GNP characterization showed that spherical 18 nm GNPs were successfully synthesized and capped with the various ligands [37,38]. Studies have shown that the interaction of GNPs with polymers impacts the size, stability, and distribution of GNPs; therefore, the particle stability was assessed in possible various cell culture conditions [38,39]. The obtained results showed that the particles were stable in cell culture environments allowing for chronic GNP-ligand exposure to the HepG2 cells.

Previously, it was found that the rate and extent of nanoparticle uptake can vary dramatically between cell lines, and therefore, uptake comparisons between different cell lines must be carefully considered [20,40]. We, therefore, only chose one cell line to compare the effect that the various GNP-ligands had on them.

In the WST-1 assay, most of the free ligands were more toxic when unbound except for BSA. The BSA-free ligand had low toxicity; however, when it was functionalized, the GNP-BSA became more toxic as the GNP-BSA concentration increased. BSA has three different domains to which cations, anions, and neutral molecules can reversibly bind and also has a free sulfhydryl group [41,42]. These binding properties could explain why the GNP-BSA became more toxic as the dose increased. This could depend on which orientation the BSA is bound to the GNP and why the low internalization resulted in high relative cytotoxicity.

The unbound citrate had high toxicity in the WST-1 assay. This shows the importance of purifying GNPs post-synthesis to prevent citrate carry-over when doing cell culture work and using GNPs synthesized with the Turkevich method. Excess citrate may lead to acidosis [43]. The GNP-citrate was non-toxic overall as it had a low amount of relative cytotoxicity, and the cells were still able to thrive and grow. They did, however, cause DNA damage which was not repaired. This could be due to the depletion of ATP due to the citrate affecting the cells’ ability to repair the damaged DNA [44]. One study investigated GNP-citrate’s effect on the metabolome and saw that 18 nm GNP-citrate lowered ATP production [45]. A different study reported that the high intake of the GNP-citrate was found as agglomerates in vesicles [46]. Fraga and colleagues found their 18 nm GNP-citrate to induce DNA damage in HepG2 cells at low concentrations [34]. In comparison, Das and colleagues found their 20 nm GNP-citrate to be both nephrotoxic and hepatotoxic [47].

GNP-GSH also had low cytotoxicity despite the high internalization percentage with the DNA damage being repaired. A study showed that 1.2 nm GNP-GSH had low cytotoxicity [48]. An interesting observation seen in the WST-1 results was that the lowest concentration for GNP-GSH had higher toxicity than the highest concentration of the treatment concentration and had the slowest recovery time in the xCELLigence assay. This could result from GSH toxicity, where the GSH depletes the natural cellular copper pool, resulting in cell death [49]. Another mechanism that may be attributed to cell death due to GSH toxicity could be the depletion of NADPH stores in the cell [50,51]. These mechanisms would need to be investigated to verify these assumptions.

GNP-MUA had a low internalization percentage with relative cytotoxicity equivalent to internalization and unrepaired DNA damage. One study found no significant oxidative stress seen in rat hepatocytes upon exposure to 15 nm GNP-MUA; however, it found them to be cytotoxic [52]. Another study of 60 nm GNP-MUA had different results as the particles were non-cytotoxic, which agreed with the previous study showing 33 nm GNP-MUA to be non-genotoxic [34]. This contradicts the findings in this study.

GNP-PEG was the third-largest molecule with a neutral charge; its internalization percentage was low and had low relative cytotoxicity. There was DNA damage evident at the 24 h time-point. Similar findings with 20 nm GNP-PEG causing genotoxicity in BEAS-2 B cells were published, stating that neither size nor functionalization could account for the genotoxic effects seen in the cell line [53]. Another article reported a low internalization of 15 nm GNP-PEG which formed cytotoxic aggregates in the cytoplasm. There was no evidence that the GNP-PEG accumulated in the nucleus or the mitochondria of the HepG2 cells [46].

The apoptotic induction time-point for each GNP-ligand appeared to be transient apoptosis; as seen in Figure 4a, a section of the impedance plateaued off, and then the impedance between the 180 min and 1440 min continued to increase. It also appeared that the IC30 concentration of the GNP-ligands had a beneficial effect on the cells. After the 1440 min interval, all the GNP-ligands (excluding GNP-GSH) impedance doubled from the 0.6 starting point with an end-point impedance of 1.2. The beneficial low levels of oxidative stress on the cells due to the GNP-ligands are a possible explanation for why the cell growth rate doubles [54].

Seeing that GNP-citrate, GNP-MUA, and GNP-PEG could not repair the DNA damage, further investigations looking at the mechanisms behind this would benefit understanding cellular-nanoparticle interactions more. Other aspects needing investigation include the integrity of the GNPs once entering the cells by using methods, such as transmission electron microscopy energy-dispersive X-ray spectroscopy (TEM-EDX), using HR-TEM to visualize the effects on particle morphology in cell culture environments, and assessing a range of non-toxic GNP-ligand concentration on the DNA damage. These are limitations in this study. Another aspect to consider would be how these 18 nm GNP-ligands affect gene regulation. Schaeublin and colleagues found that some genes were down-regulated after exposure to 1.5 nm gold nanoparticles. These genes included REV1 (recruits DNA polymerases to repair DNA), APEX1 (initiates base excision repair), MRE11 A, RAD21 and RAD51 (repairs double-stranded DNA breaks), ATM (Targets p53 in response to DNA double-strand breaks), GTSE1 (binds to p53 and shuttles it out of the nucleus for apoptosis repression).

## 5. Conclusions

In conclusion, this study aimed to determine the possible genotoxic effects of functionalized GNPs in HepG2 cells. Spherical GNPs functionalized with seven different biopolymers were assessed. This study had three various aspects to it. Phase 1 was the GNP-ligands’ synthesis, characterization, and stability assessment. Phase 2 comprised toxicity studies and treatment concentration determination, and phase 3 was the genotoxicity determination.

All the GNP-ligands were similar in size, with different charge properties, and stable in various biological environments. How the GNPs reacted biochemically was determined by their coating. Some GNP-ligands induced DNA damage with subsequent repair, while others caused irreparable damage despite the cellular proliferation advantage. More research is needed into the mechanisms of how these GNP-ligands are reacting and interfering with normal cellular functioning and which GNP characteristic contributes to this disruption. Although DNA was repaired, would repair continue to occur without DNA mutations if the cells were exposed to the GNP-ligands over successive generations, such as is the case in therapy where the cells are exposed to treatment over several months? The genotoxic effect of GNP-ligands and other nanoparticle types require more investigation in determining the impact of acute and chronic exposure at sub-toxic concentrations.

## Figures and Tables

**Figure 1 nanomaterials-12-01126-f001:**
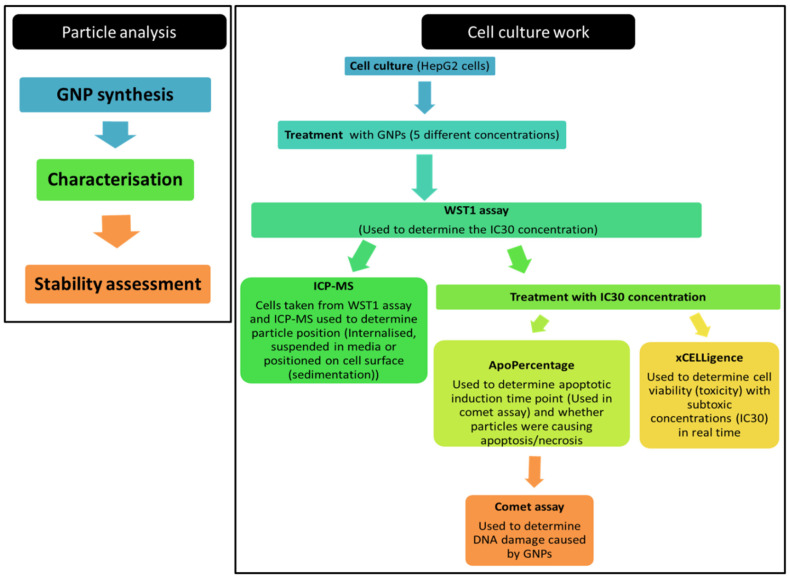
Flow diagram showing the various methods used in this study.

**Figure 2 nanomaterials-12-01126-f002:**
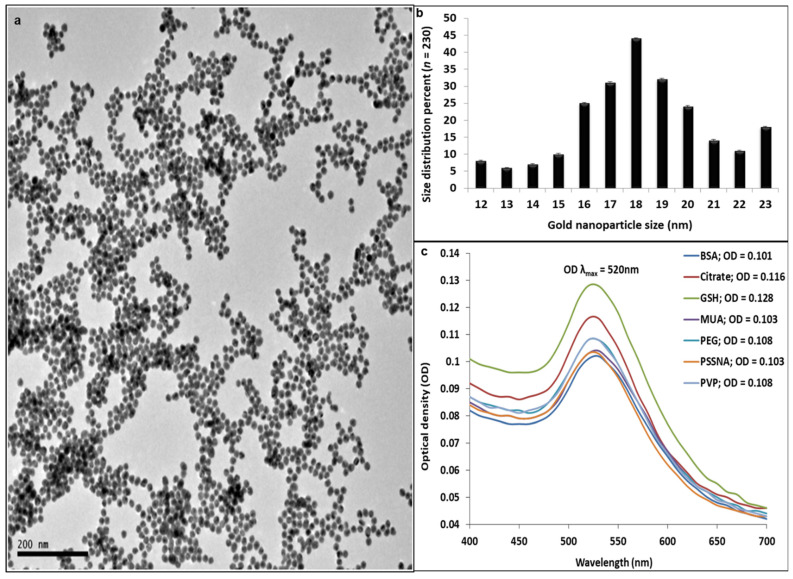
(**a**) High-resolution transmission electron microscopy micrograph of the citrate capped GNPs. (**b**) Percentage of GNP size distribution on 230 particles analyzed. (**c**) UV-Vis spectra of the functionalized GNP-ligands post sample clean-up.

**Figure 3 nanomaterials-12-01126-f003:**
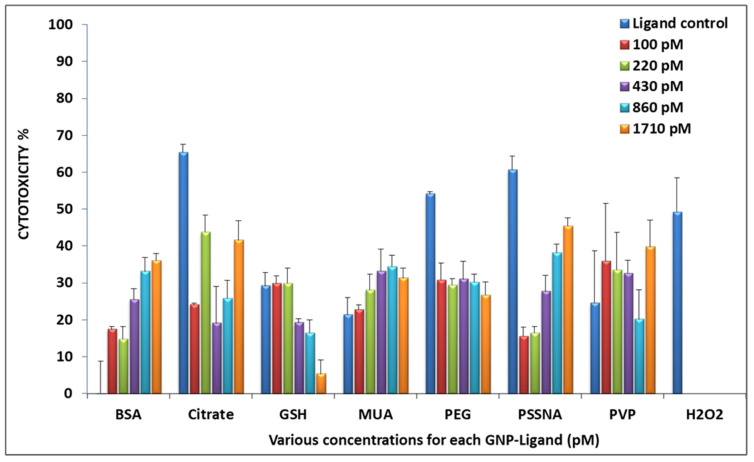
Cytotoxicity of the GNP-ligands at a range of concentrations on HepG2 cells (the ligand control contained ligand only in the same concentration as the highest concentration used, no GNP present). This data is based on an average of triplicate readings obtained for each concentration.

**Figure 4 nanomaterials-12-01126-f004:**
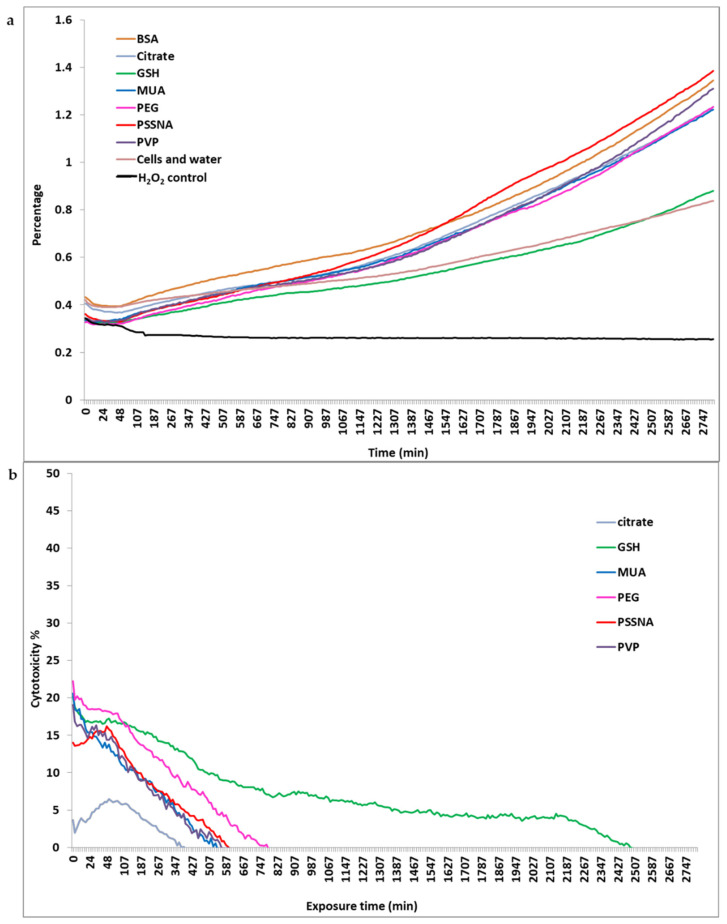
(**a**) Growth curve of the HepG2 cells after dosing them with the GNP-ligands using the IC 30 concentration determined by the impedance of the xCELLigence technology. (**b**) is the inverse graph of (**a**), where the cytotoxicity of the GNP-ligands has been determined. This data is based on an average of triplicate readings obtained for each concentration.

**Figure 5 nanomaterials-12-01126-f005:**
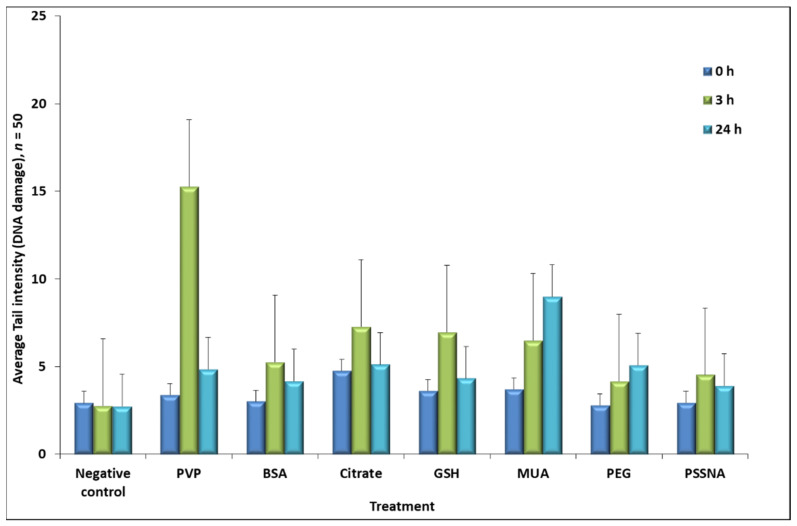
Comet assay average tail intensity for each of the GNP-ligand IC 30 concentrations at the 0 h, 3 h, and 24 h time points.

**Table 1 nanomaterials-12-01126-t001:** Summary of the GNP-ligand characteristics.

Ligand	Gel Electrophoresis: Particle Charge	DLS: Hydrodynamic Diameter (nm)
BSA	Negative	23.32
Citrate	Negative	17.40
GSH	Negative	17.33
MUA	Negative	17.21
PEG	Neutral	18.86
PSSNA	Negative	17.26
PVP	Negative	18.96

**Table 2 nanomaterials-12-01126-t002:** Cytotoxicity and cell entry of GNP-ligands.

Ligand	* WST-1	ICP-MS					ApoPercentage
	IC 30 (pM)	Cell %	Normalized IC30 (pM)	Fold Change in Cytotoxicity/Relative Cytotoxicity	PBS %	Media %	Apoptotic Time-Point
BSA	430 ± 2.96	2	9	47.8 ± 2.96	2	96	Stepwise
Citrate	860 ± 4.78	17	146	5.9 ± 4.78	3	80	Stepwise
GSH	220 ± 4.07	92	202	1.1 ± 4.07	2	6	3 h
MUA	220 ± 4.18	10	22	10.0 ± 4.18	2	88	6 h
PEG	430 ± 4.76	22	95	4.5 ± 4.76	7	71	3 h
PSSNA	430 ± 4.33	23	99	4.3 ± 4.33	0	77	3 h
PVP	430 ± 3.46	32	138	3.1 ± 3.46	2	66	3 h

* The WST−1 column is the concentration at which the IC 30 was determined. The ICP-MS column shows where most of the GNP-ligands were situated. The normalized IC 30 is calculated based on cell internalization and WST-1 concentration. The ApoPercentage column is where the apoptotic induction time-point occurred.

**Table 3 nanomaterials-12-01126-t003:** Overall percentage of DNA damage induced by the GNP-ligands.

Ligand	Time-Point at Which DNA Damage Was Determined ^#^			
	0 h	3 h	24 h	Welch *p*-Value
BSA	26	36	24	0.004 *
Citrate	24	36	36	0.494
GSH	20	38	24	0.381
MUA	24	28	46	0.003 ***
PEG	22	26	32	0.033 ***
PSSNA	20	32	28	0.103
PVP	22	48	28	0.005 *^,^**
Positive control	100	100	100	1
Negative control	12	12	12	1

^#^ DNA damage is given as a percentage of the comet tail DNA. Approximately 50 comets were analyzed per treatment for each time point. * Significant difference between 0 and 3 h. ** Significant difference between 3 and 24 h. *** Significant difference between 0 and 24 h.

**Table 4 nanomaterials-12-01126-t004:** Comparison of GNP-ligand physical parameters and their effect on HepG2 cells.

Ligand	DLS Size (nm) ± 2	Physio-Logical Charge	Cell Internalization %	Fold Change in Cytotoxicity/Relative Cytotoxicity	Apoptosis Induction Time	Cytotoxicity % (xCELLigence)	DNA Damage	DNA Repair
BSA	23.32	Negative	2	50.0	Stepwise	0	Yes	Yes
Citrate	17.4	Negative	17	5.9	Stepwise	4	Yes	No
GSH	17.33	Negative	92	1.1	3 h	20	Yes	Yes
MUA	17.21	Negative	10	10.0	6 h	18	Yes	No
PEG	18.86	Neutral	22	4.5	3 h	22	Yes	No
PSSNA	17.26	Negative	23	4.3	3 h	14	Yes	Yes
PVP	18.96	Negative	32	3.1	3 h	17	Yes	Yes

## Data Availability

Publicly archived datasets analyzed or generated during the study are stored in the Figshare repository https://doi.org/10.6084/m9.figshare.16917499 (access online, 24 March 2022).

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
