# Peer review of "A Comparison of the Genotoxic Effects of Gold Nanoparticles Functionalized with Seven Different Ligands in Cultured Human Hepatocellular Carcinoma Cells"

_nanomaterials, 2022, doi:10.3390/nano12071126_

Round 1
Reviewer 1 Report
Though the topic is not novel, the manuscript has some merit but also major pitfalls. The main goal of the study was to evaluate the in vitro toxicity of seven gold nanoparticles (GNP) surface-modified with bovine serum albumin (BSA), poly(sodium 4-styrenesulfonate) (PSSNA), trisodium citrate (citrate), mercaptoundecanoic acid (MUA), glutathione (GSH), polyvinylpyrrolidone (PVP) or polyethyleneglycol (PEG) in human hepatocarcinoma HepG2 cells. For that purpose, changes in cell viability (as assessed by the WST-1 reduction test), GNP internalisation (by ICP-MS determination of the cell’s and extracellular medium gold content), apoptosis (by the ApoPercentage assay), real-time cell growth and cytotoxicity (using the xCELLigence technology) and DNA damage (by the alkaline comet assay). Some data on the physicochemical characterization and stability of the tested GNP is also provided.
The manuscript needs an in-depth restructuring, editing and proofreading to improve the readability and clarity of the text.
What is the novelty of this study? What is the research hypothesis? What was the rational for the choice of the tested concentrations and selection of the cell model? What is the expected progress beyond the state-of-the-art?
Title
The title does not entirely reflect the content of the manuscript and should be revised.
Abstract
It should be stated in the abstract which capping agents were used for surface-modification of the GNP.
Terminology and Units
What do authors mean by “physiochemical properties”? Do they mean “physicochemical properties”? Are these two terms synonyms?
Please replace the term “solution” by “suspension” when referring to GNP.
Dosage refers a specific amount, number, and frequency of doses over a specific period of time. Please replace the term “dosage” by “concentration”.
Please use “h” and not “hr” as hour abbreviation.
Introduction
This section needs to be improved. What is the current state-of-the-art regarding the toxicity of surface-modified GNP in liver cells?
Page 1, Lines 37-40: “GNPs are being considered biologically inert as they do not get absorbed by the body, however, do accumulate in the liver as well as in cancer cells, with smaller particles excreted via the urine [13-16].”
This is controversial statement. What do authors mean by GNP not being absorbed? While is true that GNP in the blood are retained by the organs of the reticuloendothelial system (e.g liver and spleen), GNP do not exhibit a preferential accumulation in cancer cells. This sentence should be revised, and its meaning clarified.
Pages 1-2, Lines 43-45: “Some of these conflicting conclusions could be attributed to GNP coating, size, physiochemical characteristics and cell lines [20].”
Are the coating and size not physicochemical characteristics?
Overall, the Material and Methods are poorly described, not providing enough details to allow replication of the experiments and require considerable revision.
All materials and instruments should be identified, including the supplier’s name and location. It should be clear from this section how all of the data in the Results section were obtained.
Additionally, the supplementary material should also be revised. In some cases, the material provided is excessive, but most importantly, not adequate or a repetition of what have been already stated in the M&M.
Some examples:
“A rapidly stirring 100mL 0.25mM HAuCl4(aq) [22]”. What does the reference refer to?
The GNP concentration used for DLS measurements must be indicated. What parameter has been measured using this technique?
What was the dispersion medium for the physicochemical characterization measurements?
Only data (SD is missing, only mean values are provided!) for citrate-GNP are provided.
1x PBS ? 0.01x EDTA?
Please provide penicillin and streptomycin final concentrations in cell culture medium.
Some subsections can be merged. For instance: “Cell Biology” truly means “Cell Culture”; 2.2.1, 2.2.11, 2.2.1.2 can be merged.
“The reading obtained by the machine is a 10x dilution and therefore this value must be corrected to obtain the cell number in the cell suspension”.
This kind of statement does not add anything important to allow the method replicability.
The authors must indicate cell density value, preferably in number of cells/area unit used to seed cells for each type of plate, i.e for each type of assay.
Page 4, Lines 157-158: “The 5 dosages had final ligand concentrations of 1.71nm, 0.86nm, 0.43nm, 0.22nm and 0.10nm respectively. “
Do authors refer to the capped GNP or to the capping agent concentration? Do authors mean nM? In the graphs, concentrations are expressed in pM. Is this right? Please clarify.
Page 4, Lines 158-160: “The positive cell control contained cells with water only and the negative cell control contained cells dosed with 5mM hydrogen peroxide and a cell control of 2.2nm ligand only with no GNP.”
Page 4, Line 167 “Cell control = untreated cells in medium and water.”
For each assay, please clarify how negative and positive control cells have been exposed and provide the concentrations used, when applicable. For each assay, indicate how many independent experiments have been performed and the number of replicates/experiment.
Comet assay
Please indicate plating density.
“The 0hr was to see if the dosage caused immediate DNA damage.”
To achieve this goal, a control group corresponding to cells exposed only to cell culture (wo GNP) should have been included.
“The samples were neutralized with 450µl supplemented medium, transferred into autoclaved microtubes and placed back in the incubator for 1hr recovery time.”
What does this mean?
The “Real-time cell growth and cytotoxicity detection” and of the “Statistical analysis” descriptions are rather confusing.
Results
The authors do not provide any data on the physicochemical characterization in incubation medium (cell culture medium?)
Only a TEM image for citrate-GNP is provided.
WST-1 data
Data should be plotted by GNP and not by concentration. This way the reader will have a better picture of the concentration effect for each type of surface-modified GNP. Ideally, citrate-GNP should be the first in the sequence since all the others were obtained from citrate-GNP and data can be compared relative to citrate-GNP.
The ligand controls (cells incubated with the capping agents used; the concentrations of the capping agents tested must be indicated) have a significant reduction of the cell viability, which is surprisingly considering that, according to the authors, all the capping agents are biocompatible. What is the justification for this finding? This somehow compromise the study.
Table 1 is difficult to understand. Are the values presented means? The authors must also indicate the corresponding deviation values. I would recommend providing a separate table or figure with the internalization data.
Table 2. Are the values presented means? The authors must also indicate the corresponding deviation values.
Is is advisable to test more than one non-toxic concentration when assessing DNA damage by the comet assay. It would have been nice to see a GNP concentration-dependent DNA damaging effect. As it is presented, no solid conclusions about GNP DNA damaging ability can be drawn.
Author Response
Dear Reviewer, thank you very much for your valuable comments. I believe that it has contributed greatly to improving the manuscript.
|
Reviewer’s headings |
Comment |
Response |
|
Initial intro questions |
What is the novelty of this study? |
The novelty of this study is that it compares the toxicity and genotoxicity of differentially functionalized gold nanoparticles with 7 accepted ligands. Several toxicity studies have been done on some of the ligands and very limited studies on others. |
|
What is the research hypothesis? |
Different ligands of functionalized gold nanoparticles may or may not result in the genotoxicity of HepG2 cells. |
|
|
What was the rational for the choice of the tested concentrations and selection of the cell model? |
The tested concentrations were based on the IC 30 which allows for the cell to still be functional with enough toxicity to exhibit a change in the cellular biochemistry. HepG2 was chosen as it is a common cell line used in GNP toxicity which would aid in drawing a parallel of results obtained in this article to those found in the literature. |
|
|
What is the expected progress beyond the state-of-the-art? |
I am not quite sure if I understand this question clearly, but after this work, the progress would contribute towards understanding the toxicity of GNPs. This would set a platform for more toxicity investigations thus growing the GNP toxicity knowledge which will help other researchers who are designing therapies, treatments or detection methods using GNPs in designing safer products. |
|
|
Title |
The title does not entirely reflect the content of the manuscript and should be revised. |
This was changed to “A comparison of the genotoxic effects of gold nanoparticles functionalized with seven different ligands in cultured human hepatocellular carcinoma cells” |
|
Abstract |
It should be stated in the abstract which capping agents were used for surface-modification of the GNP. |
This was added to the abstract |
|
Terminology and units |
What do authors mean by “physiochemical properties”? Do they mean “physicochemical properties”? Are these two terms synonyms? |
physicochemical is dependent on the joint action of both physical and chemical processes while physiochemical is of or pertaining to both physiology and chemistry. I double-checked the context of the use of “physiochemical” in this article to make sure that it was being used in the correct context. It is the correct word in our case as we are looking at what effect the GNPs have on cell viability, toxicity, genotoxicity. These fall under physiology. However, if the reviewer feels strongly about this point we would be happy to adjust it accordingly. |
|
Please replace the term “solution” by “suspension” when referring to GNP. |
This was replaced in multiple areas throughout the manuscript. |
|
|
Dosage refers a specific amount, number, and frequency of doses over a specific period of time. Please replace the term “dosage” by “concentration”. |
Dosage was replaced by concentration |
|
|
Please use “h” and not “hr” as hour abbreviation. |
This was corrected in the manuscript as well as in the result tables. |
|
|
Introduction |
This section needs to be improved. What is the current state-of-the-art regarding the toxicity of surface-modified GNP in liver cells? |
This was added to the manuscript “GNPs are considered relatively biologically inert. Depending on their size they are excreted via the urine or may accumulate in the liver. There are many in vivo studies that have indicated how nanomaterials induce toxicity, however, these studies are limited by short time intervals and fail to consider the regeneration abilities of the liver. An example of GNPs causing toxicity in hepatocytes includes excessive production of free radicals and reactive oxygen species (ROS); which results in carbonylation of cellular proteins, DNA damage, and lipid peroxidation with more devastation linked to the smaller sizes.” |
|
Page 1, Lines 37-40: “GNPs are being considered biologically inert as they do not get absorbed by the body, however, do accumulate in the liver as well as in cancer cells, with smaller particles excreted via the urine [13-16].” This is controversial statement. What do authors mean by GNP not being absorbed? While is true that GNP in the blood are retained by the organs of the reticuloendothelial system (e.g liver and spleen), GNP do not exhibit a preferential accumulation in cancer cells. This sentence should be revised, and its meaning clarified. |
This sentence was rewritten to make the meaning clearer. |
|
|
Pages 1-2, Lines 43-45: “Some of these conflicting conclusions could be attributed to GNP coating, size, physiochemical characteristics and cell lines [20].” Are the coating and size not physicochemical characteristics? |
In this context, it is physiochemical because it is related to how these particles are interacting with the cell (its external cellular characteristics or cellular biochemistry etc.) |
|
|
Overall, the Material and Methods are poorly described, not providing enough details to allow replication of the experiments and require considerable revision. |
This section was improved with the extra steps explained in the supplementary information added to the main text. |
|
|
All materials and instruments should be identified, including the supplier’s name and location. It should be clear from this section how all of the data in the Results section were obtained.
|
This was added to the manuscript as well as a paragraph about where the reagents were purchased from. |
|
|
Additionally, the supplementary material should also be revised. In some cases, the material provided is excessive, but most importantly, not adequate or a repetition of what have been already stated in the M&M.
|
All the duplicated supplementary information was removed. |
|
|
Some examples |
“A rapidly stirring 100mL 0.25mM HAuCl4(aq) [22]”. What does the reference refer to? |
This reference was moved to the mention of the Turkevich method. |
|
The GNP concentration used for DLS measurements must be indicated. What parameter has been measured using this technique? |
More information was added to the DLS section, including what was being measured and the volume of particles added to water for appropriate machine dilution. |
|
|
What was the dispersion medium for the physicochemical characterization measurements? |
||
|
Only data (SD is missing, only mean values are provided!) for citrate-GNP are provided. |
Two of the other reviewers had a similar comment, in stating that it is mentioned in the article that statistical analysis was done, however, statistical analysis was only done on the comet assay data for DNA damage. The WST-1 data was used to determine which GNP-ligand dosage caused 30% cytotoxicity and this dosage was the dosage used for further analysis. The statistical information was rewritten to clarify this in the manuscript. |
|
|
1x PBS ? 0.01x EDTA? |
All abbreviations were explained in the manuscript. |
|
|
Please provide penicillin and streptomycin final concentrations in cell culture medium. |
The final concentrations were 10 % FBS and 1 % Penicillin-streptomycin. This was made clearer in the manuscript. |
|
|
Some subsections can be merged. For instance: “Cell Biology” truly means “Cell Culture”; 2.2.1, 2.2.11, 2.2.1.2 can be merged. |
This section was merged under cell culture work |
|
|
“The reading obtained by the machine is a 10x dilution and therefore this value must be corrected to obtain the cell number in the cell suspension”. This kind of statement does not add anything important to allow the method replicability. The authors must indicate cell density value, preferably in number of cells/area unit used to seed cells for each type of plate, i.e for each type of assay. |
This was removed from the manuscript.
Cell number/area was added with it being 7500 cells/0.32cm2 well. This density was used for all the assays using a 96-well plate. This was also clarified in the manuscript. |
|
|
Page 4, Lines 157-158: “The 5 dosages had final ligand concentrations of 1.71nm, 0.86nm, 0.43nm, 0.22nm and 0.10nm respectively. “ Do authors refer to the capped GNP or to the capping agent concentration? Do authors mean nM? In the graphs, concentrations are expressed in pM. Is this right? Please clarify. |
The nM concentration mentioned in the text was converted to the pM concentration to match the graph. We were meaning the capping agent concentration assuming that functionalization was 100%. This was clarified in the manuscript. |
|
|
Page 4, Lines 158-160: “The positive cell control contained cells with water only and the negative cell control contained cells dosed with 5mM hydrogen peroxide and a cell control of 2.2nm ligand only with no GNP.” Page 4, Line 167 “Cell control = untreated cells in medium and water.” For each assay, please clarify how negative and positive control cells have been exposed and provide the concentrations used, when applicable. For each assay, indicate how many independent experiments have been performed and the number of replicates/experiment. |
The controls for each assay were written more clearly. The number of replicates for each independent experiment was added to the methods section as well.
|
|
|
Comet assay |
Please indicate plating density. |
This was elaborated in a more detailed version of the materials and methods section of the corrected manuscript. |
|
“The 0hr was to see if the dosage caused immediate DNA damage.” |
This was the negative control stated in Table 2. This was also clarified in the manuscript. |
|
|
To achieve this goal, a control group corresponding to cells exposed only to cell culture (wo GNP) should have been included. |
||
|
“The samples were neutralized with 450µl supplemented medium, transferred into autoclaved microtubes and placed back in the incubator for 1hr recovery time.” What does this mean? |
This is to reduce any damage which may be caused by handling which could possibly lead to inaccurate results. The recovery time gives the cells time to recover if any damage did occur. |
|
|
The “Real-time cell growth and cytotoxicity detection” and of the “Statistical analysis” descriptions are rather confusing. |
These descriptions were adjusted for more clarity. |
|
|
Results |
The authors do not provide any data on the physicochemical characterization in incubation medium (cell culture medium?) Only a TEM image for citrate-GNP is provided. |
TEM was only done for the citrate-GNPs as they were the model particle. The functionalization process is via ligand exchange where the citrate is replaced by the other ligands. This is seen by the UV-Vis spectra that the particles were still spherical, the size characteristic table was also added to the result at the beginning of the article. No characterization was done with the particles in the cell culture medium and this was added as a limitation to this study in the discussion section. |
|
WST-1 data |
Data should be plotted by GNP and not by concentration. This way the reader will have a better picture of the concentration effect for each type of surface-modified GNP. Ideally, citrate-GNP should be the first in the sequence since all the others were obtained from citrate-GNP and data can be compared relative to citrate-GNP.
|
It was also suggested by other reviewers that all the data follows an alphabetical format to make it easier for the readers, thus, this was corrected right throughout the manuscript. The WST-1 data was, however, clustered according to each type of surface-modified GNP but kept alphabetically. |
|
Table 1 is difficult to understand. Are the values presented means? The authors must also indicate the corresponding deviation values. I would recommend providing a separate table or figure with the internalization data.
|
The normalized IC 30 is calculated based on what percentage of the GNP-ligands was internalized at the specific IC 30 concentration. Fold change in cytotoxicity / relative to cytotoxicity considers the cytotoxicity of the GNP-ligand concentration based on the percentage of GNP-ligands internalized. This was added to the manuscript to clarify table 1.
The normalized IC 30 was calculated by: Cell% and WST-1 concentration. E.g. for BSA = 2/100*430 = 9pM
Fold Change in cytotoxicity / Relative cytotoxicity was calculated by the WST-1 concentration/Normalized IC e.g. 430/9 = 47.8
The standard deviation was added to the table and the raw data uploaded to the repository. |
|
|
Table 2. Are the values presented means? The authors must also indicate the corresponding deviation values.
|
DNA damage is given as a percentage of the comet tail DNA. Approximately 50 comets were analyzed per treatment for each time point. No, it is not the mean, it was the number of comets that had damaged out of the 50 analyzed for that specific time point and treatment. |
|
|
It is advisable to test more than one non-toxic concentration when assessing DNA damage by the comet assay. It would have been nice to see a GNP concentration-dependent DNA damaging effect. As it is presented, no solid conclusions about GNP DNA damaging ability can be drawn.
|
This is a preliminary study, and this comment was also added in the discussion as future consideration when investigating the GNP-ligands which had no DNA repair. |
Reviewer 2 Report
Journal: Nanomaterials
Manuscript ID: nanomaterials-1580027
Type of manuscript: Article
Title: The genotoxic effect of gold nanoparticles in cultured human cells
Authors: Danielle Mulder, Cornelius Johannes Francois Taute, Mari van Wyk and Pieter J. Pretorius
In this manuscript submitted as Article to Nanomaterials, the authors reported very interesting and useful results of a well designed and conducted study on synthesis, characterization, internalization and genotoxic effects of small gold nanoparticles functionalized with different ligands. The topic of this manuscript is relevant to the field of the Nanomaterials and fits with the scope of this journal. The paper has a good structure, is very well written (apart from a few grammar or typing errors) and concise, and the style captivates the reader. The methodology is described in detail, and the results are very clear presented. The conclusions are based on the obtained results. Despite the 3 minor aspects that have to be corrected (and that could be solved in parallel with the editing process), this reviewer recommends the publication of the manuscript into the Nanomaterials journal.
Minor points:
1. The authors should provide the full origin of the reagents, apparatuses and software used (company, city, [state], country).
2. In Figure 3b instead of PVP should be PSSNA for the blue line (the legend within the panel b);
3. Some examples of grammar or typing errors:
- p.2 in 2.2.1., line 66: please correct the name of Erlenmeyer;
- p.2 in 2.2.2, line 73: please reformulate “The seven ligands (coating agents) chosen for functionalization was chosen based on”. Also change the number of the paragraph into 2.1.2.;
- p.3 in 2.1.5, line 118: please reformulate “The various parameters investigated were typically used biological buffers at”;
- p.5 in 2.2.2.1, lines 153-154: please reformulate “7500 cells/well were seeded in a 96-well plate with 100μL medium and overnight attach-ment in culturing conditions”;
- p.10 in 3.3., line 331: please reformulate “In Table 2 contains the overall percentage of DNA damage”;
- p.11 in 4., lines 363 and 366: please be consistent with the form of the used word (dependant/dependent);
- p.11 in 4., lines 377-378: please reformulate “Another study reported that the high intake of the GNP-citrate were found as agglomerates in vesicles”;
4. There is an inconsistency in formatting of 28 references in the References list.
Author Response
Dear Reviewer, thank you very much for your valuable comments. I believe that it has contributed greatly to improving the manuscript.
|
Reviewer’s points |
Comment |
Response |
|
1 |
The authors should provide the full origin of the reagents, apparatuses and software used (company, city, [state], country). |
This was added for all the reagents and apparatuses mentioned in the manuscript. |
|
2 |
In Figure 3b instead of PVP should be PSSNA for the blue line (the legend within the panel b); |
This was corrected and the colors representing the results for the various ligands are now the same for figure 3a and b. |
|
3 |
Some examples of grammar or typing errors: |
All the grammar and spelling were corrected. The South African spelling was also converted. e.g. Foetal bovine to fetal bovine. |
|
|
- p.2 in 2.2.1., line 66: please correct the name of Erlenmeyer; |
|
|
- p.2 in 2.2.2, line 73: please reformulate “The seven ligands (coating agents) chosen for functionalization was chosen based on”. Also change the number of the paragraph into 2.1.2.; |
||
|
- p.3 in 2.1.5, line 118: please reformulate “The various parameters investigated were typically used biological buffers at”; |
||
|
- p.5 in 2.2.2.1, lines 153-154: please reformulate “7500 cells/well were seeded in a 96-well plate with 100μL medium and overnight attach-ment in culturing conditions”; |
||
|
- p.10 in 3.3., line 331: please reformulate “In Table 2 contains the overall percentage of DNA damage”; |
||
|
- p.11 in 4., lines 363 and 366: please be consistent with the form of the used word (dependant/dependent); |
||
|
- p.11 in 4., lines 377-378: please reformulate “Another study reported that the high intake of the GNP-citrate were found as agglomerates in vesicles”; |
||
|
4 |
There is an inconsistency in formatting of 38 references in the References list. |
This was corrected. |
Reviewer 3 Report
The present work studied the genotoxic and cytotoxic effects of gold nanoparticles in combination with different ligands against Human hepatocellular carcinoma HEPG2 cells.
The purpose of the work is clearly defined. Materials and methods are described in detail. The results are also neatly presented and are discussed in detail.
There are a few minor comments aimed at improving the text of the manuscript.
1) In the Abstract
The authors have to decipher all used abbreviations. Please, write " Human hepatocellular carcinoma HEPG2 " on line 11. Please do it for other abbreviations as well as you made it on line 7 for GNPs.
2) Materials and Methods
Line 62. Please, provide a reference.
Please provide for all used equipment the following information: company, town, country.
3) Results
Table 1. Please, show statistics.
Table 2. Please, describe what is shown in Table 2. Are the average values shown?
Author Response
Dear Reviewer, thank you very much for your valuable comments. I believe that it has contributed greatly to improving the manuscript.
|
Reviewer’s points |
Comment |
Response |
|
1) In the Abstract |
The authors have to decipher all used abbreviations. Please, write " Human hepatocellular carcinoma HEPG2 " on line 11. Please do it for other abbreviations as well as you made it on line 7 for GNPs. |
All the abbreviations were explained in the abstract and the rest of the manuscript. |
|
2) Materials and Methods |
Line 62. Please, provide a reference. |
The correct reference was moved to line 62 (Ref #22). |
|
Please provide for all used equipment the following information: company, town, country. |
This was added to the manuscript as well as a paragraph about where the reagents were purchased from. |
|
|
3) Results |
Table 1. Please, show statistics. |
Two of the other reviewers had a similar comment, stating that it is mentioned in the article that statistical analysis was done, however, statistical analysis was only done on the comet assay data for DNA damage. The WST-1 data was used to determine which GNP-ligand dosage caused 30% cytotoxicity and this dosage was the dosage used for further analysis. The statistical information was rewritten to clarify this in the manuscript. The standard deviation was added to the table. |
|
Table 2. Please, describe what is shown in Table 2. Are the average values shown? |
DNA damage is given as a percentage of the comet tail DNA. Approximately 50 comets were analyzed per treatment for each time point. No, it is not the mean, it was the number of comets that had damaged out of the 50 analyzed for that specific time point and treatment. |
Reviewer 4 Report
In this paper, Mulder and co-workers have presented the results on effects of differently coated gold nanoparticles on cell viability and genotoxicity of HepG2 cells. A lot of work has been done and the study is very complex. The obtained results are interesting but their current presentation in the manuscript is rather confusing and needs to be modified. Moreover, some additional information should be included.
Major remarks
I.
The major flaws of this manuscript are in presentation of data obtained on characterisation and stability analyses of golden nanoparticles coated with 7 different ligands. For each type of the ligand we need to see data on their morphology, size and charge upon synthesis and upon their addition to the exposure medium. These data need to be clearly presented at the beginning of the Results section. Therefore, the following needs to be done
- Authors should provide data on GNPs analyses obtained by TEM for all types of GNPs! They have presented TEM photo only for citrate-capped GNPs?! What about GNPs with other coatings? Adequate TEM photos should be provided for GNPS with all tested coatings to be able to see their shape and morphology.
- In 2.1.4.1. UV-Vis Spectrometry – why is SPR used for size determination of only citrate-capped GNPs? What about GNPs with other types of ligands? How was their size determined? The size of each type of GNPs analysed in this study should be presented at the beginning of the Results section, not at the end.
GNP-ligand cellular internalisation was obtained with ICP-MS. However, this method can give an information only about the presence of gold within the cells, but you cannot obtain information about the form of the gold. How can you be sure that after uptake, gold remained in the form of nanoparticles? Nanoparticles could be dissolved in the exposure medium and/or in the cells – in that case you will record the effects which at least partially can be ascribed to gold ions or the ligand itself. To avoid this, two things need to be done:
- GNP stability analyses in medium used for cell exposure – authors state that the supplementary Table 5 contains data on GNPs stability but in this table we cannot find any data how their size and charge changed in the exposure medium. All we can see in this table is Yes or No, but based on what data? Authors need to provide experimental data, not just their conclusion!
- Use TEM-EDX to detect particles within the cell (TEM part) and to confirm that these particles are actually gold (EDX part); with ICP-MS you can only detect if the gold is up taken, but you cannot detect if gold nanoparticles were taken up! That is why the use of TEM-EDX is mandatory for all 7 type of GNPs
- Btw, where we can see the results of ICP-MS analyses? What was the concentration or amount of gold that was up taken by cells upon exposure to each type of GNPs?
II.
In my opinion all details dealing with experimental procedure from the supplementary file should be included in the main manuscript. Supplementary file should only contain additional results.
Flow diagram (currently presented as M1: Overview of methods flow diagram in Supplementary materials) should be incorporated in the manuscript as Figure 1.
Since you have tested 7 different coatings, I would suggest that you keep an alphabetical order of their appearance in each tested paragraph. So far, you have a different order of coating appearance in almost each tested parameter, which makes it rather confusing for the reader and difficult to follow!
Figure 2. - Cytotoxicity results presented on Figure 2 – which results are statistically significant? Authors state that they have performed statistical analyses but we do not see results of statistical analysis!
Figure 3. – It is unclear why the effect of one GNP type (PSSNA) is presented in one way (Figure 3a) and the effects of two other types of GNPs (PVP and citrate) in another way, while the GNP-GSH is presented in both Figures?
Moreover, legend within the Figure 3b suggests that the effects of GNP-BSA are also presented as dark yellow line but this line cannot be seen!
And what about two additional types of ligands (MUA and PEG); why their effects are not presented at all on Figure 3?
I find graphs presented at Supplementary Figure 7 much more informative and complete that those presented on Figure 3 in the manuscript; therefore, I would suggest currently present Figure 3 to be replaced with graphs from supplementary Figure 7.
Minor remarks
English language should be improved. There are many grammatical errors.
Values and units should be separated with one space in the whole manuscript.
There are many abbreviations which are not explained in the text.
All other specific comments can be found in the pdf of the manuscript.

Author Response
Dear Reviewer, thank you very much for your valuable comments. I believe that it has contributed greatly to improving the manuscript.
|
Comment |
Response |
|
Major remarks |
|
|
I. |
|
|
The major flaws of this manuscript are in presentation of data obtained on characterisation and stability analyses of golden nanoparticles coated with 7 different ligands. For each type of the ligand we need to see data on their morphology, size and charge upon synthesis and upon their addition to the exposure medium. These data need to be clearly presented at the beginning of the Results section. Therefore, the following needs to be done |
|
|
TEM was only done for the citrate-GNPs as they were the model particle. The functionalization process is via ligand exchange where the citrate is replaced by the other ligands. This is seen by the UV-Vis spectra that the particles were still spherical, the size characteristic table was also added to the result at the beginning of the article. No characterization was done with the particles in the cell culture medium and this was added as a limitation to this study in the discussion section. |
|
The UV-Vis size, the charge, and the hydrodynamic diameter were all added to Table 1 at the beginning of the results section. |
|
GNP-ligand cellular internalisation was obtained with ICP-MS. However, this method can give an information only about the presence of gold within the cells, but you cannot obtain information about the form of the gold. How can you be sure that after uptake, gold remained in the form of nanoparticles? Nanoparticles could be dissolved in the exposure medium and/or in the cells – in that case you will record the effects which at least partially can be ascribed to gold ions or the ligand itself. To avoid this, two things need to be done: |
|
|
The stability was assessed using UV-Vis at various time points. When the OD max of the GNP-ligand was lower than 70% of the control then it was considered unstable. These spectra were added to the repository as there are many spectra. |
|
This is a preliminary study, and this comment was added in the discussion as future consideration when investigating the GNP-ligands which had no DNA repair. |
|
The raw data was added to the expository. The ICP-MS was the number of gold atoms that were measured. This was clarified in the manuscript. |
|
II. |
|
|
In my opinion all details dealing with experimental procedure from the supplementary file should be included in the main manuscript. Supplementary file should only contain additional results. |
The extra information regarding the experimental procedure which was explained in detail was moved from the supplementary to the main manuscript. |
|
Flow diagram (currently presented as M1: Overview of methods flow diagram in Supplementary materials) should be incorporated in the manuscript as Figure 1. |
This figure was moved to the main manuscript |
|
Since you have tested 7 different coatings, I would suggest that you keep an alphabetical order of their appearance in each tested paragraph. So far, you have a different order of coating appearance in almost each tested parameter, which makes it rather confusing for the reader and difficult to follow! |
This was adjusted for all the tables and figures |
|
Figure 2. - Cytotoxicity results presented on Figure 2 – which results are statistically significant? Authors state that they have performed statistical analyses but we do not see results of statistical analysis! |
Statistical analysis was only done on the comet assay data for DNA damage. The WST-1 data was used to determine which GNP-ligand dosage caused 30% cytotoxicity and this dosage was the dosage used for further analysis. The statistical information was rewritten to clarify this in the manuscript. |
|
Figure 3. – It is unclear why the effect of one GNP type (PSSNA) is presented in one way (Figure 3a) and the effects of two other types of GNPs (PVP and citrate) in another way, while the GNP-GSH is presented in both Figures? |
Initially, Figure 3 in the manuscript only showed the GNP-ligands which had different trends to highlight the trends more easily for the reader. Then the more comprehensive graphs were added to the supplementary. This has, however, been changed. The figure now contains all the results. The colors chosen for the various GNP-ligands in Figure 3a now match that of Figure 3b to reduce confusion. The GNP-BSA legend of Figure 3b was also removed as the line is not visible on the graph with it recovering too quickly. |
|
Moreover, legend within the Figure 3b suggests that the effects of GNP-BSA are also presented as dark yellow line but this line cannot be seen! |
|
|
And what about two additional types of ligands (MUA and PEG); why their effects are not presented at all on Figure 3? |
|
|
I find graphs presented at Supplementary Figure 7 much more informative and complete that those presented on Figure 3 in the manuscript; therefore, I would suggest currently present Figure 3 to be replaced with graphs from supplementary Figure 7. |
|
|
Minor remarks |
|
|
English language should be improved. There are many grammatical errors. |
This was corrected |
|
Values and units should be separated with one space in the whole manuscript. |
This was corrected |
|
There are many abbreviations which are not explained in the text. |
This was corrected and all abbreviations are now explained in the text. |
|
All other specific comments can be found in the pdf of the manuscript. |
Thank you for your effort and for being thorough. These suggestions were also added and corrected in the manuscript. |
Round 2
Reviewer 1 Report
Despite some improvements, the manuscript have several flaws that compromise the present study and its conclusions.
It is not clear how is surface-capped GNP concentration calculated and expressed. The calculations presented in the supplementary materials are very confusing. It seems that surface capped-GNP concentrations are based on the ligand final concentration and not in terms of Au mass, which is rather uncommon and make the comparison with other studies difficult. Nevertheless, considering that testing concentrations are in the pM range, a significant reduction of the cell viability has been observed in cells exposed to the ligands themselves, which is surprisingly considering that, according to the authors, all the capping agents are biocompatible.
Another serious flaw concerns the way the comet assay has been carried out and how data is presented.
After cell detachment, cells were returned to the incubator for a recovery period of one hour. This procedure is not usual. At the same time, if the “0h” experimental condition was intended to see if exposure to GNP causes an immediate DNA damage, this recovery time might compromise this goal.
It is not clear what was the descriptor used to score DNA damage (% tail intensity?). How many independent experiments and how many replicates/experiments have been performed? The authors must present the % tail intensity mean values and the corresponding deviation values.
Author Response
Dear Reviewer,
Thank you very much for your valuable feedback and assessment of the article. It is greatly appreciated. The GNP concentration and comet assay sections were extensively edited for readership clarity. The article was also extensively language edited and over 400 grammar and sentence corrections were made to the article. Please see the table below for the corrections made to the article based on your concerns.
|
Comment |
Response |
|
It is not clear how is surface-capped GNP concentration calculated and expressed. The calculations presented in the supplementary materials are very confusing. It seems that surface capped-GNP concentrations are based on the ligand final concentration and not in terms of Au mass, which is rather uncommon and make the comparison with other studies difficult. Nevertheless, considering that testing concentrations are in the pM range, a significant reduction of the cell viability has been observed in cells exposed to the ligands themselves, which is surprisingly considering that, according to the authors, all the capping agents are biocompatible. |
The diameter and concentration of the GNPs were calculated as ±18 nm and 1.99 nM, respectively, using the approach and equation from Haiss et al. The section of the particle characterization in the methods section and the results section was rewritten to make more sense and cause less confusion for the readers.
It was also stated in the WST-1 assay methods section that the “The 5 concentrations had final ligand concentrations of 1710 pM, 860 pM, 430 pM, 220 pM, and 100 pM, respectively. These calculations assume that all the ligands were functionalized onto the GNP surface. “
Even though the ligands were used in biological applications, the dosage of the ligand-only control was based on the highest dosage of the GNP-ligands. This is not the dosage that would be used for therapy or treatment. Any reference to them being biocompatible was removed from the article to avoid confusion of it being a therapy concentration. |
|
Another serious flaw concerns the way the comet assay has been carried out and how data is presented. After cell detachment, cells were returned to the incubator for a recovery period of one hour. This procedure is not usual. At the same time, if the “0h” experimental condition was intended to see if exposure to GNP causes an immediate DNA damage, this recovery time might compromise this goal.
It is not clear what was the descriptor used to score DNA damage (% tail intensity?). How many independent experiments and how many replicates/experiments have been performed? The authors must present the % tail intensity mean values and the corresponding deviation values. |
The section under the comet assay methods and results were amended for clarity purposes.
To address the recovery time comment this was added to the article “Cells were grown and sub-cultured into a 24-well plate then treated with the IC 30 concentrations of the respective GNP-ligands. An hour recovery time was implemented before the slide preparation step. This recovery time reduces the possible adverse effects on the cell handling procedure, which may skew the results. A pitfall of this recovery time is that if the GNP-ligand did cause immediate DNA damage (at 0 h), the cells could repair it during that incubation time; however, the focus was on the longer-term damage that survived the incubation time.”
Yes, the descriptor used to score the DNA damage was the % tail intensity, and the experiment was done in triplicate, with a total of 50 comets per GNP-ligand and time point were analyzed. This information was also added to the manuscript.
Figure 5 was added to include the % tail intensity mean values vs treatment.
|
Reviewer 4 Report
Authors have adequately modified their article which is now suitable for publication.
Author Response
Dear Reviewer,
Thank you very much for your feedback.
Kind regards